METHODS AND RESOURCES

# GALDAR: A genetically encoded galactose sensor for visualizing sugar metabolism *in vivo*

**Uğurcan Sakizli**[1,2], **Tomomi Takano**[1,3], **Sa Kan Yoo**[1,2,3]*

**1** Laboratory for Homeodynamics, RIKEN BDR, Kobe, Japan, **2** Division of Developmental Biology and Regenerative Medicine, Kobe University, Kobe, Japan, **3** Physiological Genetics Laboratory, RIKEN CPR, Kobe, Japan

* sakan.yoo@riken.jp

**Data Availability Statement:** All relevant data are within the paper and its Supporting Information files.

**Funding:** This work was supported by AMED-PRIME (17939907), the JSPS KAKENHI

## Abstract

Sugar metabolism plays a pivotal role in sustaining life. Its dynamics within organisms is less understood compared to its intracellular metabolism. Galactose, a hexose stereoisomer of glucose, is a monosaccharide transported via the same transporters with glucose. Galactose feeds into glycolysis and regulates protein glycosylation. Defects in galactose metabolism are lethal for animals. Here, by transgenically implementing the yeast galactose sensing system into *Drosophila*, we developed a genetically encoded sensor, GALDAR, which detects galactose *in vivo*. Using this heterologous system, we revealed dynamics of galactose metabolism in various tissues. Notably, we discovered that intestinal stem cells do not uptake detectable levels of galactose or glucose. GALDAR elucidates the role for galactokinase in metabolism of galactose and a transition of galactose metabolism during the larval period. This work provides a new system that enables analyses of *in vivo* sugar metabolism.

## Introduction

Sugar metabolism is central for life. Glycolysis is a main metabolic pathway, which provides macromolecules and can feed into the oxidative phosphorylation pathway [1]. Monosaccharides such as glucose, galactose, and fructose can enter glycolysis. Although metabolic pathways within cells have been elucidated, metabolism *in vivo* is still enigmatic. For example, lactate was recently discovered as an important circulating metabolite that functions as a primary energy source for peripheral tissues, contrary to the long-held belief that glucose plays the role [2]. A recent study suggests that fructose generated from glucose is important for sensing glucose [3]. These exemplify that our knowledge of organismal sugar metabolism is still limited.

We focused on galactose, which is known for its importance in prokaryotic ecology, mammalian development, and *Drosophila* physiology [4–7]. Galactose was identified in milk by Louis Pasteur in 1856 [4]. Since the molecular structures of galactose and glucose are similar, differing in their stereochemistry at carbon 4, the same transporters, SGLTs (Na$^+$-D-glucose

(JP16H06220, JP22H02807) and JST FOREST (JPMJFR216F) to S.K.Y. The funders had no role in study design, data collection and analysis, decision to publish, or preparation of the manuscript.

**Competing interests:** U.S. and S.K.Y. are co-inventors on a patent application based on this work filed by RIKEN.

costransporters) or GLUTs transport them in all known cases [8]. Consistently, mutations in SGLT1 gene are responsible for glucose–galactose malabsorption in humans [9]. Galactosemia, which is caused by defects of galactose metabolism, is a lethal disease for animals.

We decided to develop a genetically encoded galactose sensor *in vivo* because (1) galactose metabolic dynamics within organisms remains to be solved; and (2) by suppressing galactose metabolism, we could use a galactose sensor as a proxy of glucose/galactose uptake measurement. The endogenous galactose amount is expected to be low, which makes the flux analyses sensitive, contrasting with the robust maintenance of the intracellular glucose concentrations. We also note that compared to availability of genetically encoded sensors for glucose [10–12], there has been no biosensor developed for galactose. Although the lab's standard fly food does not contain much galactose, marula, an African fruit that is the ancestral host of *D. melanogaster* contains galactose [13,14] and many fruits and vegetables contain both free and bound galactose [15–17], which suggests that galactose is a physiologically relevant sugar for *Drosophila*. In fact, adult flies can keep living with only galactose (S1A Fig), indicating that *Drosophila* uses galactose as an energy source. Galactose metabolism genes such as *galactokinase*, *GALT* (Galactose-1p uridyltransferase), and *GALE* (UDPgal 4'-epimerase) are phylogenetically conserved, including *Drosophila* [6]. Importantly, mutants of these enzymes accumulate galactose-derived metabolites and demonstrate shortened lifespan, a lower climbing ability and a fecundity defect, which are aggravated by addition of extra galactose [6,18–23]. These indicate that galactose is a physiologically and ecologically relevant and evolutionarily conserved monosaccharide for flies as well as vertebrates. Although previous studies clarified importance of galactose metabolism in *Drosophila*, intracellular dynamics of galactose *in vivo* remains unclear. Here, taking advantage of the yeast endogenous galactose sensing system, we generated an *in vivo* galactose sensor, which we named GALDAR.

## Results

### Incorporation of the yeast Gal system into *Drosophila*

We heterologously incorporated the galactose sensing system of yeast into *Drosophila*. Yeast detects galactose using Gal4, Gal80, and Gal3. In absence of galactose, Gal80 binds to the transcriptional activator Gal4 and inhibits transcription of genes with a UAS sequence [24,25]. Gal4 sequestration by Gal80 is lifted upon a conformational change of galactose-bound Gal3, which has a high affinity to Gal80, allowing the transcriptional machinery to be recruited by free Gal4 and consequent transcription [24,25]. The Perrimon group introduced the Gal4-UAS system to *Drosophila* [26], followed by addition of Gal80 by the Luo group [27]. However, galactose sensing Gal3 has never been incorporated to *Drosophila*. We transferred the whole Gal system of yeast to *Drosophila* and generated a heterologous biosensor for galactose.

We introduced the yeast galactose metabolism genes, including Gal3, Gal4, and Gal80 together with a fluorescent protein driven under the UAS sequence into *Drosophila* through a transgenic approach. We named this sensor GALDAR (GALactose raDAR). In the first version of GALDAR, GALDAR1 (G1), Gal3, and Gal80 proteins separated by a self-cleaving P2A sequence [28] were ubiquitously expressed utilizing a tubulin enhancer (Fig 1A). G1 can be used as a tissue specific sensor with an appropriate Gal4 driver. In this study, we used ubiquitous Gal4 drivers, *act-Gal4* or *tub-Gal4* with G1. While developing G1, we also considered the possibility that imbalanced amounts of Gal4 and Gal80 proteins in the cell might theoretically affect results. To address this potential problem, we generated the second version of the sensor, GALDAR2 (G2). In this version, all Gal4, Gal3, and Gal80 proteins are expressed using a single transcript, separated with self-cleaving P2A sequences, making sure that each of the products

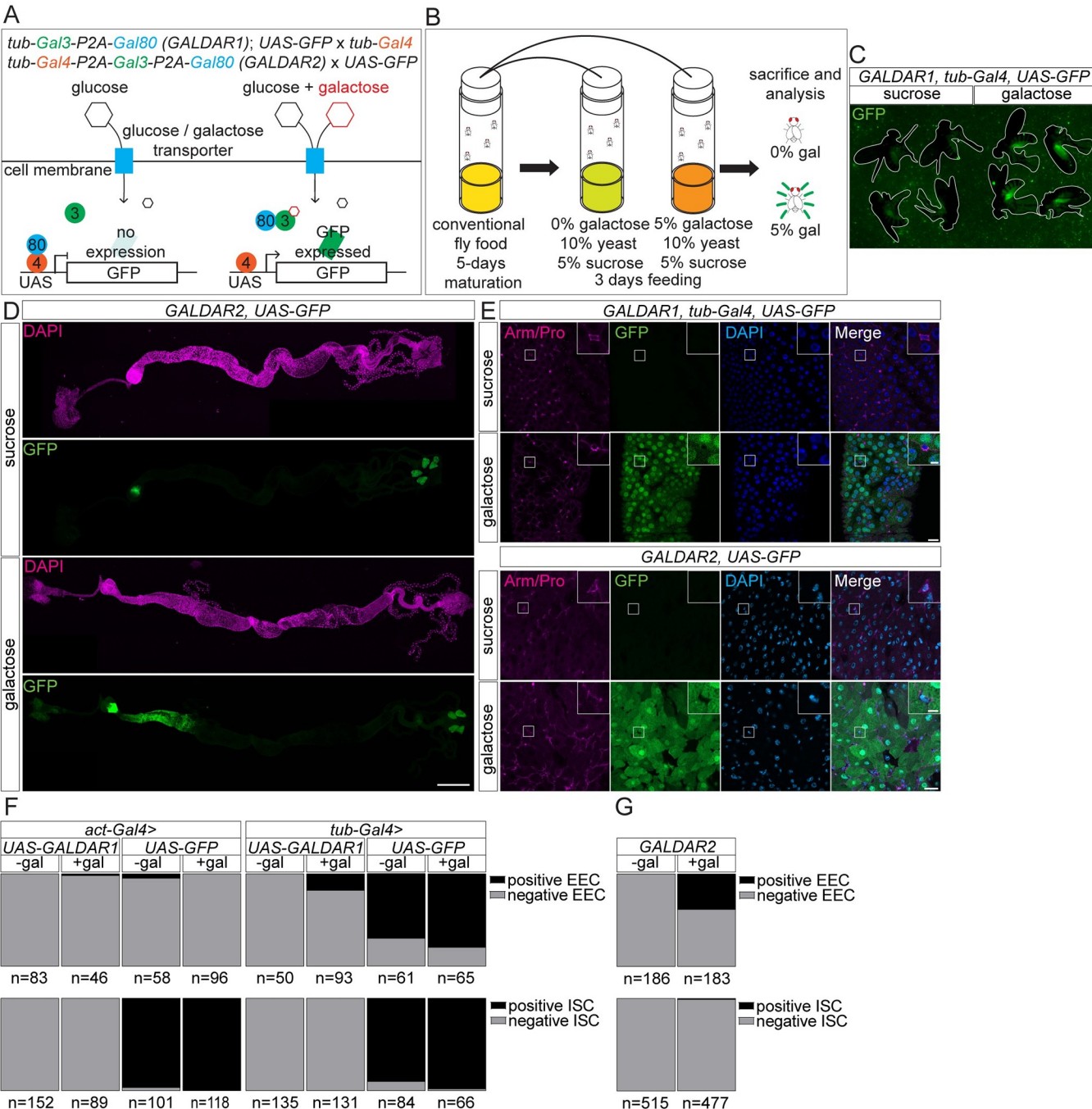

**Fig 1. GALDAR reflects intracellular galactose amounts.** (A) Schematic representation of GALDAR system. GFP is expressed only in the presence of galactose. (B) Experimental setup for analyzing GALDAR using galactose. (C) Image of whole adult female *G1*, *tub-Gal4*, *UAS-GFP* flies with (right) or without (left) galactose feeding for 3 days. Flies are outlined with white lines. (D) Tiled stacks of confocal images of whole adult midguts. Anterior midguts of *G2*, *UAS-GFP* flies are GFP positive in a galactose dependent manner. (E) Confocal images of R2 regions of *G1*, *tub-Gal4* and *G2* midguts with *UAS-GFP*. Examples of ISCs (Armadillo+, Prospero-) are highlighted in upper-right panels in each picture. (F) Quantification of *G1*, *UAS-GFP* and *UAS-GFP* positive EECs (upper panel), and ISCs (lower panel) using *act-Gal4* (left) and *tub-Gal4* (right). (G) Quantification of *G2*, *UAS-GFP* positive EECs (upper panel) and ISCs (lower panel). Data are representative of at least 3 independent experiments. *n* = number of cells in (F) and (G) from at least 5 individual samples. S1 Data provides the source data used for all graphs and statistical analyses. Scale bars, 500 μm in (D), 20 μm in large, and 5 μm in small panels in (E).

is generated at relatively similar amounts (Fig 1A). The readout of both G1 and G2 sensors are the same: without galactose no signals from fluorescent proteins such as GFP observed while with galactose fluorescence signals appear (Fig 1A).

To allow time for maturation of the midgut [29], ingestion of food, and transcription, translation and maturation of fluorescent proteins, we fed 5-day-old virgin female *GALDAR* flies for 3 days with galactose supplemented and control food (Fig 1B). After 3 days, the flies fed with galactose supplemented food exhibited GFP signals, whereas control flies exhibited no visible GFP signals (Fig 1C). Since the midgut is the first tissue for absorption and metabolism of consumed foods, we analyzed the midgut. We observed that the anterior midgut (R2 region) of adult virgin females exhibits GFP signals with a presence of galactose (Fig 1D). This observation is reminiscent of the previous reports showing that expression of maltase and polysaccharide metabolism-related genes is compartmentalized in the anterior midgut of female *Drosophila* [30,31]. Since we observed similar patterns of GALDAR signals in R2 of the midgut in mated females and males (S1B and S1C Fig), we used virgin females in this study.

We characterized cell types in the midgut using G1 and G2 sensors. Enterocytes are the main cell type positive for GALDAR as expected, since they are the absorptive cells in the midgut. Intriguingly, using G1 (*act-Gal4*, *tub-Gal4*) and G2, we found that GFP signals were absent in ISCs, indicating that ISCs have very little amount of galactose (Fig 1E–1G). Regarding enteroendocrine cells (EECs), we noticed that *act-Gal4* itself does not promote expression of *UAS-GFP* in EECs, whereas *tub-Gal4* promotes expression of *UAS-GFP* in approximately 80% of analyzed EECs (Figs 1E, 1F, and S1D). Thus, for analyzing galactose in EECs, *act-Gal4* cannot be used while *tub-Gal4* is usable with a caveat that Gal4 is expressed in a majority of, but not all EECs. With *tub-Gal4*, approximately 20% of analyzed EECs were positive for GALDAR with galactose, suggesting a subpopulation of EECs maintains the galactose concentration high (Fig 1F and 1G). We also confirmed that, due to Gal3's specificity for galactose, GALDAR is not activated by feeding sucrose or glucose (S1E and S1F Fig).

Outside of the gut epithelium, visceral muscle cells wrapping around the midgut in R2 were positive for GALDAR in a galactose-dependent manner (Fig 2A). These observations indicate that enterocytes and the visceral muscles in the anterior midgut maintain the galactose concentration high (Fig 2B).

We assessed galactose levels in additional tissues besides the midgut using GALDAR. We observed that heart muscles are GALDAR positive (Fig 2C and 2D). Interestingly, the visceral muscles around the hindgut were GALDAR positive but enterocytes of the hindgut were not. Unexpectedly, we did not observe GFP signals in the fat body either. Although not statistically significant, we observed a trend of slight increase of GFP signals in oenocytes (Fig 2C and 2D). We considered a possibility that some tissues might express GFP transiently during 3-day-feeding of galactose and might lose signals at the time of the analysis. In order to exclude this possibility, we examined GFP signals at 3 different time points with a 1-day interval after starting galactose feeding. After 2 days of feeding 5% galactose, the visceral muscle around the hindgut and heart muscle signals reached a peak and plateau (S2A and S2B Fig). We conclude that the experimental condition that we used is suitable for our analyses. These observations suggest that various types of tissues exhibit varying concentrations of intracellular galactose.

## Injection of galactose to the hemocoel

The observation that metabolic tissues such as the fat body remains negative for GALDAR after galactose feeding intrigued us. One possible explanation for the lack of signals in the fat body is that the amount of galactose in the hemolymph is not high enough after feeding to generate a meaningful signal. In order to test this, we injected a galactose solution in varying

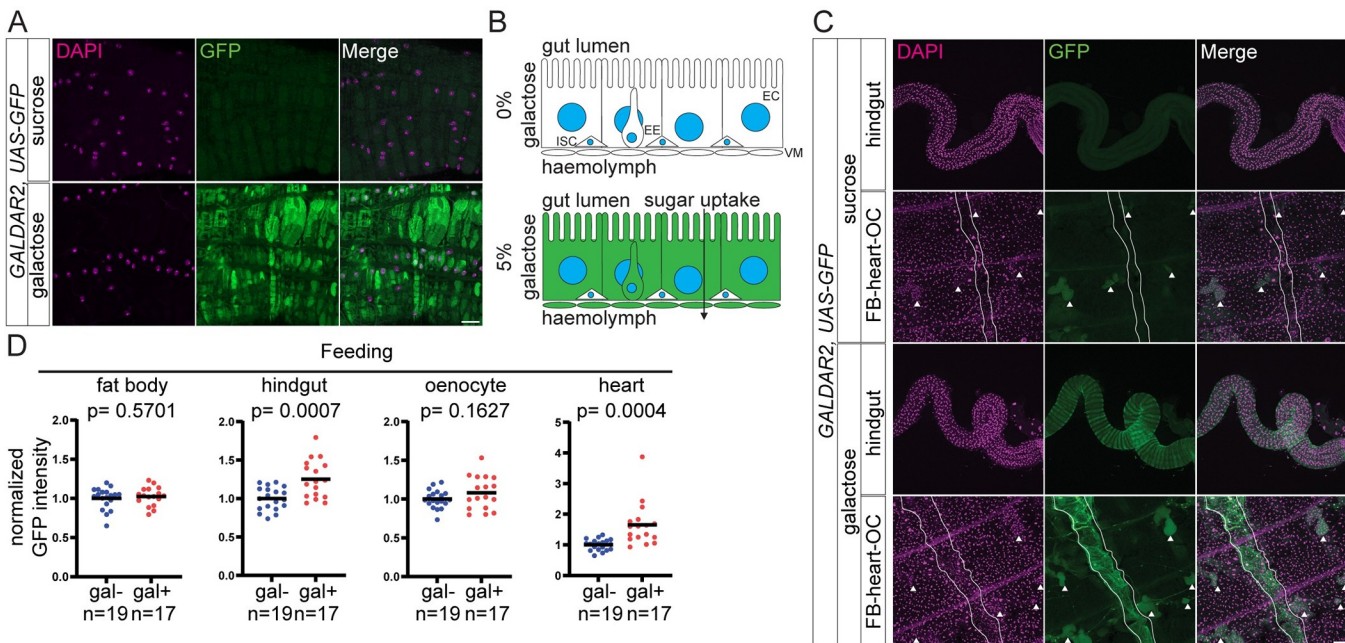

**Fig 2. GALDAR signals in relevant tissues.** (A) Confocal images of visceral muscle layer of R2 midgut. (B) Schematic representation of cells with a high galactose concentration in midgut. (C) Stacked confocal images of hindgut and abdomen of adult flies (*G2, UAS-GFP*). Visceral muscle around the hindgut and heart muscle cells are GFP positive upon galactose feeding. Arrowheads indicate oenocytes and the white lines outline the heart muscle. (D) Quantification of GFP intensity in fat body, hindgut, oenocyte, and heart muscles with or without galactose feeding, normalized to 0% galactose feeding. Data are representative of at least 3 independent experiments. *n* = number of tissues analyzed in (D). S1 Data provides the source data used for all graphs and statistical analyses. Scale bars, 20 μm in (A), 50 μm in (C).

concentrations directly into the hemocoel of adult flies and observed GFP signals 1 day later (Fig 3A). Indeed, we observed a clear increase in the signals in the fat body in a dose-dependent manner (Fig 3B and 3C). Unexpectedly, even though there is a slight increase, GALDAR signals in the heart muscle were not induced by injection as strongly as by feeding (Fig 3B and 3C), suggesting that the route of galactose incorporation affects galactose metabolism in the heart. When injected with galactose, the visceral muscle around the hindgut becomes GALDAR positive, resembling the feeding case. In contrast to feeding, enterocytes in the hindgut also became GALDAR positive, which implies an outwards flow of galactose from the hemolymph to the hindgut lumen (Figs 3B, 3C, and S2C). The lack of GALDAR signals in the Malpighian tubules even after injection of galactose at high concentrations and feeding of 5% galactose suggests that removal of excess sugar in the hemolymph might be regulated by the hindgut. We also compared GALDAR signals with feeding and injection in various tissues. Fitting feeding data to injection data revealed that, signals after feeding 5% galactose for 3 days induced GFP signals that are comparable to 1 day after injection of 5% galactose in the fat body, the hindgut and oenocytes, whereas the increase in heart muscle GFP expression was comparable to 20% galactose injection (Fig 3D). These data show that galactose signals in the fat body can be visualized with GALDAR using suitable concentrations of galactose solution, that galactose metabolism in the heart depends on the route of galactose incorporation and that there is a possible route for excess sugar excretion through the hindgut.

## Galactose metabolism through Galk affects GALDAR signals

Up to this point, we have elucidated galactose levels in various types of cells using GALDAR. Two factors can affect intracellular galactose concentrations: uptake and metabolism.

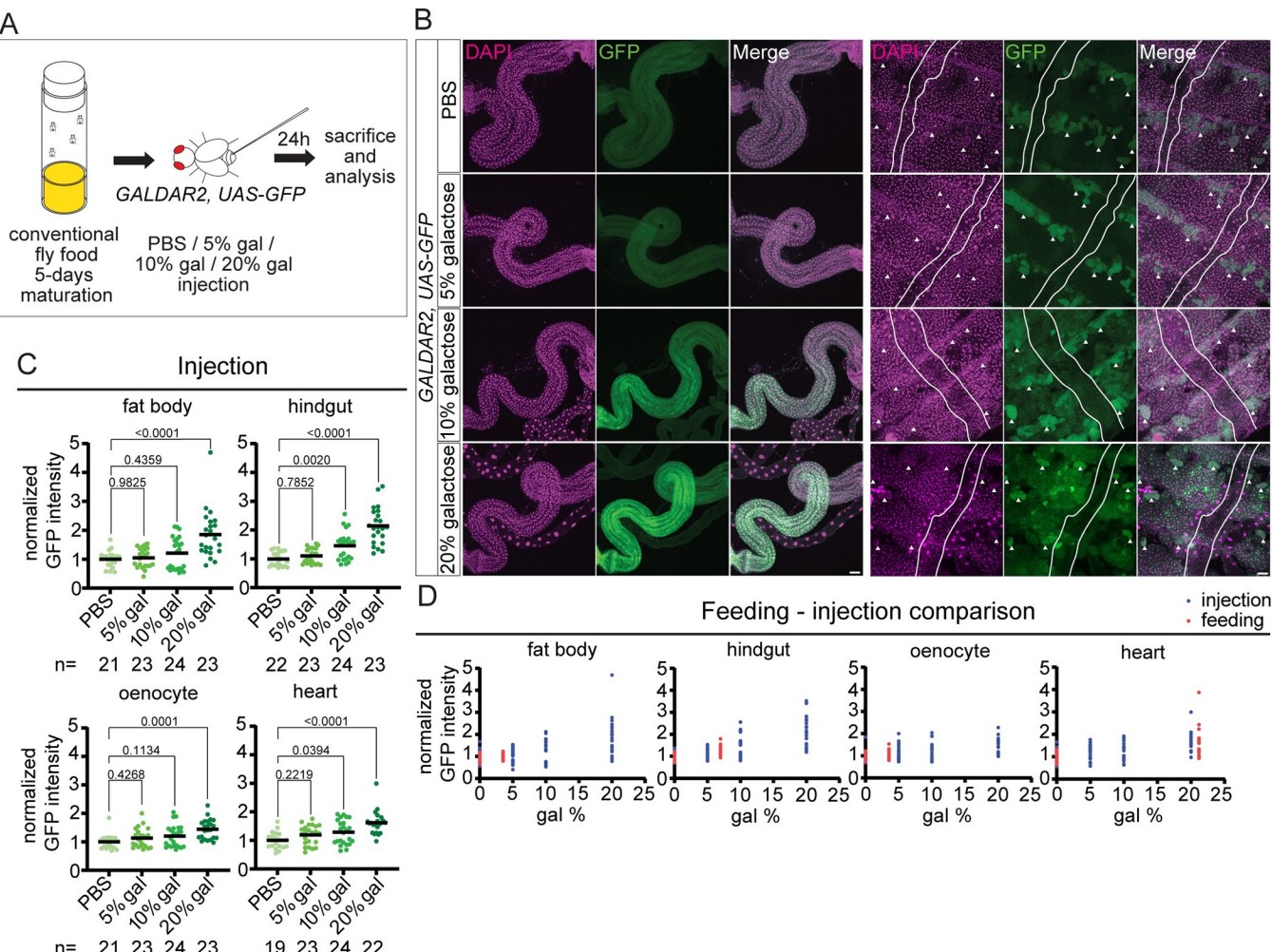

**Fig 3. Galactose injection into hemocoel.** (A) Experimental setup for galactose injection and analysis of GALDAR signals. (B) Stacked confocal images of hindgut and abdomen (heart: between white lines, oenocytes: arrowheads) of adult flies (*G2, UAS-GFP*) 1 day after galactose injection. GFP expression increases in the visceral muscle of hindgut and fat body in a dose-dependent manner. (C) Quantification of GFP intensity in fat body, hindgut, oenocyte, and heart muscle, normalized to PBS. (D) Comparison of the increase in GFP intensity using feeding (red) and injection (blue) data. Data are representative of at least 3 independent experiments. *n* = number of tissues analyzed in (C) and (D). S1 Data provides the source data used for all graphs and statistical analyses. Scale bars, 50 μm.

GALDAR reflects intracellular galactose levels and does not distinguish these 2 mechanisms. For example, absence of GALDAR signals in ISCs could indicate no uptake or rapid consumption in such a pace that prevents GALDAR activation through galactose binding to Gal3.

To estimate the contribution of the 2 mechanisms, we decided to use GALDAR under inhibition of galactose catabolism. Galactose is metabolized through the Leloir pathway, which is regulated by Galactokinase (Galk), GALT and GALE (Fig 4A). We used a *Galk* loss-of-function mutant that was generated by imprecise excision of a p-element. Galk is evolutionarily conserved and expressed almost ubiquitously in the whole fly (S3A–S3C Fig). Examining GALDAR signals in the *Galk* mutant enables us to assess only the aspect of galactose uptake.

We examined GALDAR flies with the *Galk* null mutant background. *Galk* null mutant flies showed shorter lifespan, which was further aggravated with extra galactose addition (S3D Fig), which is consistent with the previous study [22]. To prevent toxicity of galactose in the mutant, we induced GALDAR using a lower galactose concentration for a shorter duration. On the

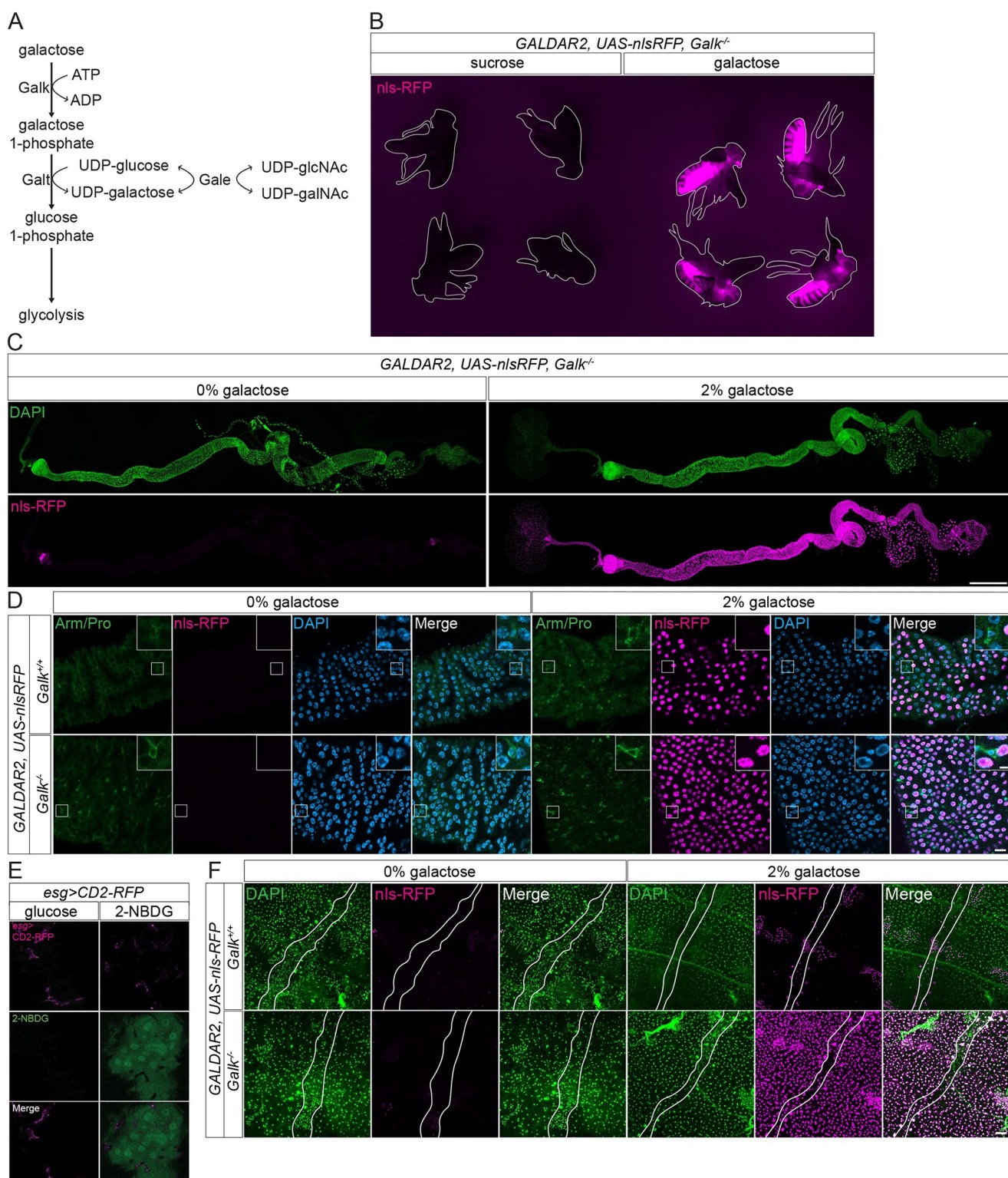

**Fig 4. Galactose metabolism affects GALDAR signals.** (A) The Leloir pathway that controls galactose metabolism. (B) Fluorescent images of whole *GALDAR2, UAS-nlsRFP, Galk⁻/⁻* whole adult virgin female flies with (right) or without (left) 2% galactose feeding for 2d (nlsRFP in magenta). Flies are outlined with white lines. (C) Tiled stacks of confocal images of *G2, nlsRFP, Galk⁻/⁻* whole adult midguts with (right) and without (left) galactose feeding for 2d. (D) Confocal images of R2 regions of *G2, nlsRFP* (top) and *G2, nlsRFP, Galk⁻/⁻* (bottom) adult midguts with (right) and without (left) 2% galactose feeding for 2d. (E) Confocal images of live adult midguts fed 10% glucose solution (left) and 10% glucose solution with 750 μm 2-NBDG (right). (F) Stacks

of confocal images of adult abdomens showing heart muscle (between white lines), fat body, and oenocytes (DAPI in green, nlsRFP in magenta). Data are representative of at least 3 independent experiments. Scale bars, 500 μm in (C), 20 μm in (D) and (E), 50 μm in (F).

*Galk* mutant background, most tissues and cell types were GALDAR positive after 2-day feeding with 2% galactose (Fig 4B). This suggests that galactose is efficiently metabolized in most tissues at a relatively high rate, which affects the intracellular concentrations of galactose detected by GALDAR. Consistently, the whole adult midgut exhibits GALDAR signals after feeding 2% galactose with the *Galk* mutant background as opposed to the restricted GALDAR signals in anterior midgut with 5% galactose feeding in the wild-type background (Fig 4C).

Interestingly, even in the background of *Galk* mutant, ISCs were GALDAR negative, indicating that ISCs do not uptake galactose (Fig 4D). Consistently, no *Galk* expression was detected in ISCs (S3A Fig). Since galactose and glucose are transported by the same transporters in all known cases, this result implies that ISCs do not uptake high amounts of glucose either. Using the fluorescent glucose analogue, 2-NBDG [32], we tested this possibility. Since 2-NBDG is water soluble, making it incompatible with immunostaining (S3E Fig), we detected 2-NBDG in a live setting. ISCs, labeled with CD8-RFP, were exclusively negative for 2-NBDG, whereas ECs were positive after feeding with the 2-NBDG solution (Fig 4E), recapitulating GALDAR's result. This indicates that ISCs cannot uptake detectable levels of glucose/galactose, although they may uptake a trace amount of it. Consistently, Fly Cell Atlas [33] indicates that ISCs have low expression of sugar transporters (S3F Fig). Similar to the midgut, fat body cells, which are GALDAR negative in the wild-type background, become positive in the *Galk* null mutant background (Fig 4F), illustrating a relatively high galactose metabolism in the fat body.

## GALDAR in development

Thus far, we have detected galactose by GALDAR in adult flies. We investigated galactose metabolism during larval development. During larval development, there is an ecdysteroid-mediated dynamic change of sugar metabolism [34–36]. GALDAR larvae were cultivated under distinct galactose concentrations, specifically 0% and 5% (Fig 5A). Initial observation of GALDAR signals in larvae suggested a visible decrease of GFP signals as larvae transition from L2 to L3 (Fig 5B). Detailed analyses of tissues indicate that while the midgut and the epidermis maintain similar GALDAR signals in L2 and L3, the fat body reduces the signals during the transition from L2 to L3, contributing to the overall reduction of GFP signals in L3 (Fig 5C).

The loss of GFP signals in the fat body during larval development might be a result of altered galactose metabolism through *Galk*, and/or its uptake. qPCR analysis using larval fat bodies showed an increase of *Galk* transcripts in L3 larvae in comparison to L2 (Fig 5D). In accordance with increased Galk expression, the galactose concentration in L3 larvae was nearly undetectable when the larvae were grown in normal food and decreased significantly when the larvae were grown in 5% galactose containing food in comparison to higher amounts in L2 in both cases (Fig 5E). Although we cannot exclude the involvement of galactose uptake, our results imply that the *Galk* expression change during the larval development might regulate galactose dynamics in the fat body (Fig 5F).

## Engineering of a transcription independent, Gal4/UAS compatible GALDAR

Until now, we demonstrated the versatility of GALDAR1/2 in detection of intracellular galactose levels using larval and adult *Drosophila*. Despite its strengths, GALDAR has 2 drawbacks. First, GALDAR's dependency on Gal4/UAS prohibits tissue-specific gene manipulation such

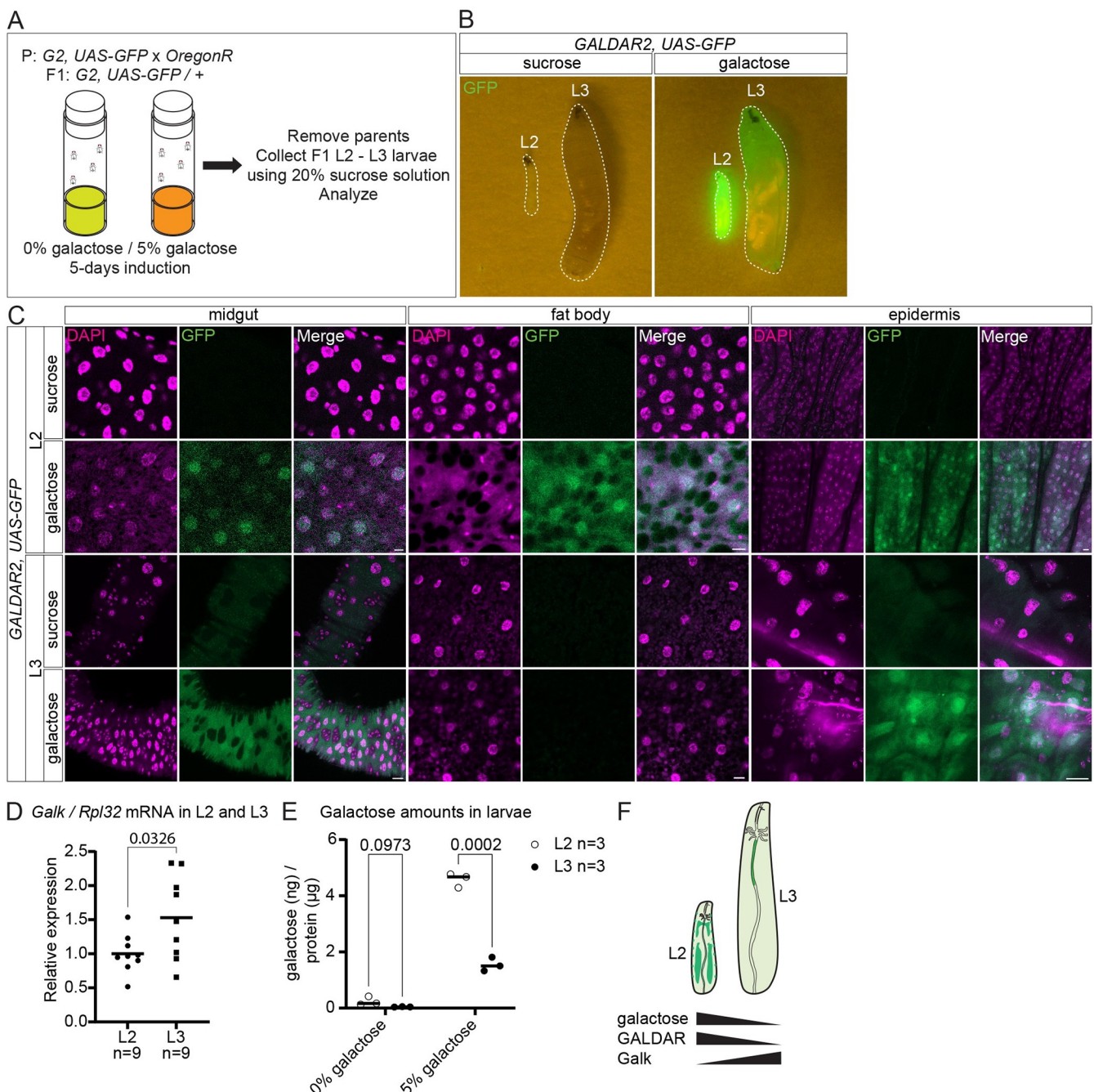

**Fig 5. Transition in galactose metabolism in larval stages.** (A) Experimental setup of analyzing GALDAR in larvae. (B) Fluorescence images of whole L2 and L3 larvae with (right) or without (left) galactose feeding. (C) Confocal images of L2 (top) and L3 (bottom) midguts, fat bodies, and epidermises (DAPI in magenta, GFP in green). (D) qPCR analysis of *Galk* expression in L2 and L3 larvae. Two-tailed unpaired *t* test. (E) Galactose amounts in L2 and L3 larvae detected by LC-MS/MS normalized by protein. (F) Proposed metabolic transition in larval stages. Data are representative of at least 3 independent experiments. S1 Data provides the source data used for all graphs and statistical analyses. Scale bars, 5 μm in all L2 images and 20 μm in all L3 images in (C).

as RNAi or ectopic expression by Gal4/UAS. Second, its temporal sensitivity to detect galactose is relatively low due to the time delay between cellular uptake of galactose and visible fluorescent protein signals, which takes time for transcription, translation, and maturation of the fluorescent protein. Although biosensing based on transcription has unique advantages such

as its ability to capture long-term sustained states and to manipulate cells with high signals [37], as exemplified by transcription-based genetically encoded calcium biosensors such as CaLexA [38] and TRIC [39], we acknowledge the low temporal resolution of GALDAR1/2.

To overcome these problems, we generated a new version of GALDAR, which we named GALDAR3 (G3). In G3, we removed Gal4 and used a mutant version of Gal80 (G310D) [24], which can interact with Gal3 normally but is defective in binding to Gal4. These changes isolated the sensor from the Gal4/UAS system. Additionally, GFP was fused to the mutant Gal80 and Gal3 was confined to the nucleus with addition of a nuclear localization signal. These proteins are expressed ubiquitously using a single transcript under the tubulin enhancer. In the absence of intracellular galactose, Gal80 remains diffused in the cell and shuttles in and out of the nucleus constantly [40], resulting in a diffused fluorescent signal. When galactose is present in cells, Gal80*-GFP interacts with nuclear Gal3 and gets trapped in the nucleus causing an observable and quantifiable increase in nuclear fluorescence intensity (Fig 6A). Since the sensor is already ubiquitously expressed, time delay between galactose uptake and sensing is minimized, making it more sensitive than G1/2.

We confirmed that G3 can detect galactose in enterocytes of the midgut and the fat body by feeding *tub-G3* flies with 0% and 5% galactose containing food for 2 days. We observed a significant increase in nuclear GFP intensities depending on galactose addition (Fig 6B and 6C). Enterocytes in both R2 and R4 as well as fat body cells had visibly higher nuclear GFP signals, showing that G3 is more sensitive and less affected by endogenous galactose metabolism. We also confirmed that G3 can be used in combination with tissue-specific Gal4/UAS. To demonstrate this, we combined G3 with enterocyte-specific overexpression of CD8-RFP using *Myo1D-Gal4*. CD8-RFP labels membranous structures in enterocytes, especially with a higher intensity in the membrane-rich apical brush border. Enterocyte-specific CD8-RFP was observed independently of the presence of galactose and G3 (Fig 6D). Nuclear GFP signals in enterocytes increased significantly with the addition of galactose (Fig 6E). These data indicate that G3 and Gal4/UAS work independently without interfering each other.

Finally, we investigated whether GALDAR can detect physiological levels of galactose in food that exists in the wild besides artificially added galactose. By design, G3 is more sensitive than G1/2, thus we used G3 for this purpose. We fed G3 flies with fig fruit (*Ficus* spp.) for 2 days. Fig plants, which contain a relatively high amount of galactose [17], are reported to be highly infested by *Drosophilidae* showing that it is a relevant food source in the wild [41]. We were able to detect significantly higher nuclear GFP intensities in the midgut enterocytes and the fat body of fig fed flies compared to 0% galactose (Fig 6F and 6G). These data demonstrate that G3 can detect intracellular galactose when flies eat natural fruits.

## Discussion

Here, we developed a new genetically encoded galactose sensor in *Drosophila* through heterologous incorporation of the yeast Gal system. We developed transcription-dependent (G1 and G2) and -independent (G3) platforms.

Using GALDAR, we elucidated spatiotemporal differences in galactose concentrations in tissues of adult and larval flies. In general, uptake and metabolism of galactose affect its intracellular concentration. G1/G2 labeled enterocytes in the anterior midgut while G1/G2 in the *Galk* mutant and G3 labeled enterocytes in both anterior and posterior parts of the midgut. Although clarification based on experimentation is necessary, this strongly suggests that the regional difference of G1/G2 signals stems from the galactose metabolism difference: the anterior region has slower metabolism and the posterior has faster one. Consistently, Fly Cell Atlas [33] indicates that enterocytes in the posterior region tend to have higher expression of genes

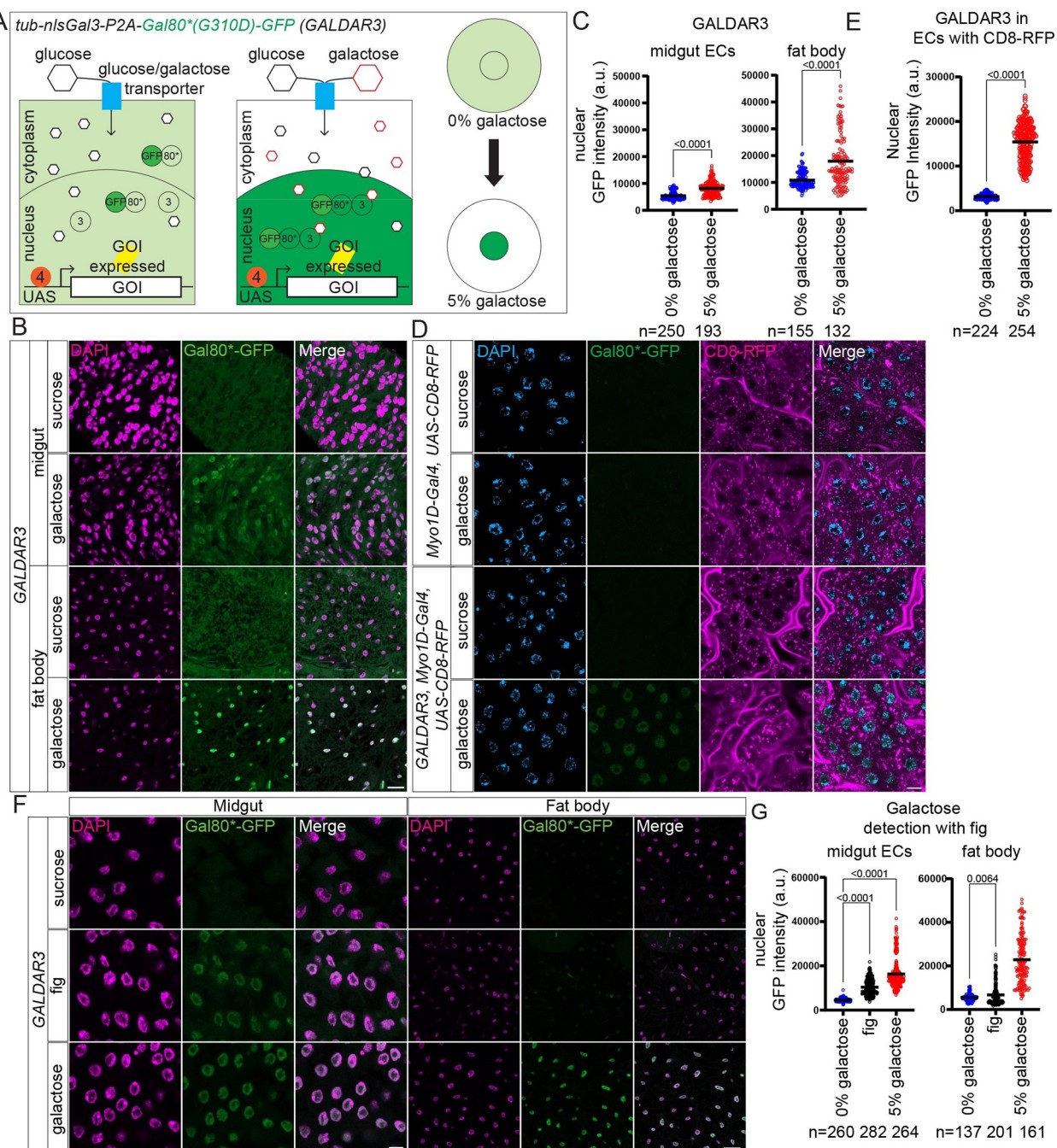

**Fig 6. Design and application of GALDAR3.** (A) Schematic representation of GALDAR3 mechanism. Upon increase in intracellular galactose amounts, Gal80*(G310D)-GFP binds to nlsGal3, increasing nuclear GFP intensity. G3 is independent of Gal4/UAS system, and any gene of interest (GOI) can be expressed normally. (B) Confocal images of R2 regions of *GALDAR3* midguts (top) and fat body (bottom) after 2 days feeding with or without galactose supplemented food (DAPI in magenta, Gal80*GFP in green). (C) Quantification of nuclear GFP intensities of midgut ECs and fat body cells. Two-tailed unpaired *t* test. (D) Confocal images of R2 regions of *Myo1D-Gal4, UAS-CD8-RFP* midguts (top) or *GALDAR3, Myo1d-Gal4, UAS-CD8-RFP* (bottom) in presence and absence of galactose (DAPI in blue, Gal80*-GFP in green, CD8-RFP in magenta). CD8-RFP pictures represent a more apical z-section of the same confocal stack in order to demonstrate signals in brush border. (E) Quantification of nuclear GFP intensities of midgut ECs in *GALDAR3, Myo1d-Gal4, UAS-CD8-RFP* flies in absence and presence of galactose. Two-tailed unpaired *t* test. (F) Confocal images of ECs in R2 regions of midguts (left) and fat body (right) of adult virgin *G3* flies fed with sucrose (top), fig (middle), and 5% galactose addition (bottom) for 2 days (DAPI in magenta, Gal80*GFP in green). (G) Quantification of nuclear GFP intensities of images in (F). Midgut ECs, ANOVA with multiple comparison test; fat body, two-tailed unpaired *t* test. Data are representative of at least 3 independent experiments. S1 Data provides the source data used for all graphs and statistical analyses. Scale bars, 20 μm in images in (B), 10 μm in (D), 10 μm in midgut and 20 μm in fat body in (F).

related to galactose and glucose metabolism than in the anterior region (S4A and S4B Fig) while expression levels of sugar transporters are comparable in both regions. This is in a contrast with the previous findings of restricted expression of maltase and polysaccharide metabolism-related genes in the anterior midgut of female flies, indicating that each process of sugar metabolism is compartmentalized in different regions of the gut. In ISCs, we experimentally demonstrated that they cannot uptake a detectable level of galactose/glucose. This finding in ISCs raises an interesting question what nutrition ISCs use for their survival, which is worthwhile for future mechanistic studies.

We postulate there are following applications of GALDAR.

First, GALDAR enables analysis of galactose metabolism *in vivo*. In spite of its importance for organisms, intracellular dynamics of galactose *in vivo* has been unclear. Due to its robustness of signals and versatility, GALDAR should be able to be applied to other organisms and mammalian tissue culture besides *Drosophila*. G3 provides a higher temporal resolution and is more sensitive than G1/2. Thus, G3 is useful to detect relatively small amounts of galactose while G1/2 is useful to detect accumulative long-term signals that can be saturated for detection by G3. For example, the regional difference of signals in the gut was more pronounced with G1/2 than G3 likely due to signal saturation with G3. Although we used *Drosophila* in this study, GALDAR can be applied to any eukaryotic organism.

Second, GALDAR can be used as a proxy of glucose uptake sensor, especially under the condition of galactokinase inhibition, since the same transporters transport galactose and glucose. By using GALDAR as an uptake sensor for glucose, we may not have to use the fluorescent glucose analog 2-NDBG or isotope labeled glucose, which has cellular toxicity or relative difficulty to use. GALDAR can be used with a single wavelength under a simple fluorescence microscope. As a proof of principle, using GALDAR, we discovered that intestinal stem cells do not uptake detectable levels of glucose/galactose. We speculate that GALDAR can be utilized as a screening platform for drugs that affect glucose uptake in mammalian tissue culture.

Third, in the tissues that uptake galactose, G1/2 can be used as a gene inducible system, which is an advantage of transcription-dependent biosensor [37]. This will be a new addition to the current inducible systems of Gal4 such as GeneSwitch with RU486 [42,43], TARGET with Gal80$^{ts}$ [44], Gal80-DD with TMP [45], and AGES with auxin [46] in the tissues where cells uptake galactose readily.

The versatility and modularity of the GALDAR platform may enable other applications besides the ones mentioned above. We envision that GALDAR can open up a new avenue of sugar metabolism research and pave the way for its multifaceted application across various domains.

## Methods

### Fly husbandry

Flies were maintained as previously described [47]. Standard fly food used for maintenance contained 0.8% agar, 10% glucose, 4.5% corn flour, 3.72% dry yeast, 0.4% propionic acid, and 0.3% butyl p-hydroxybenzoate. Crosses and experiments were conducted in 25°C incubators. Collected virgins were flipped to new food at least 3 times per week. Unless otherwise noted, 5 to 8 days virgin females and males were used for experiments.

### *Drosophila* stocks

The following stocks were used in this study:
   *Oregon R*
   *w-; act-Gal4/CyO-dfd-YFP*

*w-;; tub-Gal4/TM6B*
*w-;; UAS-GFP*
*w-; UAS-nlsRFP*
*w-; esg-LexA*
*w-;; LexAop-CD2-RFP*
*w-; Myo1D-Gal4/CyO; UAS-CD8-RFP/TM6B*
*w- tub-Gal3-P2A-Gal80 (GALDAR1)*
*w-; tub-Gal3-P2A-Gal80/CyO; UAS-GFP/TM6B*
*w-; tub-Gal4-P2A-Gal3-P2A-Gal80 (GALDAR2)*
*w-;; tub-nlsGal3-P2A-Gal80(G310D)-GFP (GALDAR3)*
*w-;; tub-Gal4-P2A-Gal3-P2A-Gal80, UAS-GFP/TM6B*
*w-; tub-Gal4-P2A-Gal3-P2A-Gal80, UAS-nlsRFP/CyO*
*Galk-Gal4 (from BDSC, 94405)*
*Galk[DeltaEXC9] (from BDSC, 93186)*
*w-; tub-Gal4-P2A-Gal3-P2A-Gal80, UAS-nlsRFP/CyO; Galk[DeltaEXC9]*

## Analysis of survival on sugars

Five to 8 days old males collected in standard food vials were flipped into vials containing no galactose addition or with 5% galactose addition to a basal food containing 1.5% agar and 0.5% sucrose. Flies were flipped every 2 days. Dead flies are noted and graphed using GraphPad Prism. For *Galk* mutants, food was renewed every day.

## Generation of GALDAR1, GALDAR2, and GALDAR3 fly stocks

DNA of Gal3-2A-Gal80 (G1), Gal4-2A-Gal3-2A-Gal80 (G2), and nlsGal3-2A-Gal80-GFP (G3) were synthesized by GenScript and were subcloned into pCaSpeR4 with a tubulin enhancer. G310D mutation of Gal80 was introduced by overlap PCR using following primers:

*Gal3 F*
*CAAGCGGCCGCCAAAATGAAC*
*Gal80 R*
*CCAGGTTACTTATTTCTGCGAAGTCTGCG*
*Gal80 F*
*CGCAGACTTCGCAGAAATAAGTAACCTGG*
*GFP R*
*CAATCTAGATTATTTATAGAGCTCATCCATACC.*
Transgenic flies with the plasmids were generated by BestGene.

## Induction of GALDAR with galactose supplementation and fig

Flies collected in standard food vials were flipped into vials containing either no galactose (1.5% agar, 10% yeast, 5% sucrose, 0.3% butyl p-hydroxybenzoate, 0.3% propionic acid) or galactose supplemented (1.5% agar, 10% yeast, 5% sucrose, 5% galactose, 0.3% butyl p-hydroxybenzoate, 0.3% propionic acid) food for 3 days. *Galk* null mutant flies and their controls are fed 2% galactose supplemented food for 2 days. For determination of signal appearance, foods containing 0%, 1%, and 5% galactose with 1.5% agar were used for 1d, 2d, and 3d induction. For experiments using larvae, homozygous *GALDAR2, UAS-GFP* virgin females and *OregonR* males were mated and larvae with 1 copy of *GALDAR2, UAS-GFP* were raised in vials containing no galactose and galactose supplemented food. For G3 measurements, the same food conditions for adults were used and flies were fed 2 days. For experiments involving fig, a piece of cut fig fruit was placed into an empty food vial and flies were fed with the fruit for 2 days. For

the analysis of activation of GALDAR with various sugars, 5% sucrose, 5% glucose, 30% glucose, and 5% galactose food was prepared in 1.5% agar and flies were fed for 3 days.

## Galactose injection into hemocoel

Roughly 200 nl solutions of PBS, 5% galactose, 10% galactose, and 20% galactose were injected into the ventral abdomen of female GALDAR2, UAS-GFP flies using a microinjector (Narishige IM-31, Tritech Research) and glass-pulled capillaries. Injected flies were put on standard fly food and analyzed 1 day later.

## Immunostaining and confocal imaging

Immunostaining was performed as previously described [48]. Midguts and ventral abdomens of adult flies containing fat body, oenocytes, and heart were dissected and fixed for 1 h at RT in 1xPBS with 4% PFA (Thermo Fisher Scientific, 43368). Samples were washed 3 times for >10 min in PBSTx (1xPBS +0.1% Triton X-100) and incubated with primary antibodies in 10% NGS (Wako 143–06561, Sigma, #G9023-10ML), PBSTx either 4 h at RT or overnight at 4˚C. After washing 3 times with PBSTx for >10 min, the samples were incubated with DAPI and secondary antibodies in 10% NGS, PBSTx either 4 h at RT or overnight at 4˚C. Finally, samples were washed 3 more times with PBSTx and mounted in Slow Fade Diamond Antifade Mountant (Invitrogen, S36963). Before mounting, parts of remaining dorsal abdomens were removed using a micro-dissecting scissors for ease of imaging. Fluorescence images were acquired with a confocal microscope (Zeiss LSM 900).

## Quantification of immunohistochemistry and fluorescence intensity

For each IHC sample and for detection of GALDAR activation with sugars, mean fluorescence intensity of stacked 15 μm thick confocal images of the tissues was measured in 3 different areas using Fiji and the average of these measurements was used as a single data point. For detection of G3 nuclear GFP intensities, the GFP fluorescence intensity in individual ROIs drawn using DAPI signal was used.

## RT-qPCR for determination of *Galk* expression in larval fat bodies

Total RNA was extracted from L2 (10 per sample) and L3 (5 per sample) fat bodies using the Maxwell RSC simplyRNA Tissue Kit (Promega) that includes DNase digestion, and 250 ng of total RNA was used for synthesis of cDNA using 5X PrimeScript RT Master Mix (Takara) with the buffer content, polymerase and $MgCl_2$ concentrations optimized by the producer. qPCR was done using the FastStart Essential DNA Green Master Mix (Roche) for *Galk* (Gene ID:39031) using *Rpl32* (Gene ID: 43573) as a reference. The final concentration of primers was 0.5 μm, and 2 μl of synthesized cDNA was used as template for qPCR in 20 μl in clear strips. PCR cycling conditions were as suggested by the kit provider and qPCR and fluorescence detection was done using a LightCycler 96 (Roche). Error bars represent SE. Following primers were used for qPCR:

*Galk Exon 2 F*
*CAACAATTCGGAGCGAATCCGG*
*Galk Exon 4 R*
*CGGGTCCACCACTCTTTGGAAGC*
*Rpl32 Exon 2 F*
*CCAGCATACAGGCCCAAGATCGTG*
*Rpl32 Exon 3 R*

*TCTTGAATCCGGTGGGCAGCATG.*

## 2-NBDG feeding and short-term live imaging

The 4d old esg-LexA, LexAop-CD2-RFP virgin female flies were starved for 5 h and fed with 10% glucose solution or 750 μm 2-NBDG (Cayman Chemical Company) in 10% glucose solution for 2 h. Midguts are dissected in PBS, embedded in 4% low melting agarose in PBS, and imaged immediately. For confirmation of loss of 2-NBDG signals, guts were imaged in the same way and subjected to IHC as explained above.

## Detection of galactose amounts in larvae using LC-MS/MS

*OregonR* parents were flipped into food containing 1.5% agar, 5% glucose, 10% yeast or 1.5% agar, 5% glucose, 5% galactose, and 10% yeast for several days. For each sample, after removal of parents 15 L2 or 5 L3 wandering larvae were isolated from the vials using 20% sucrose solution and washed in 70% EtOH 3 times in order to remove the excess food and immediately frozen at −80˚C in 1.5 ml Eppendorf tubes. The aqueous phase was collected from the frozen samples and was centrifuged and dried in a vacuum concentrator as previously described (Sasaki and colleagues). Each dry sample was dissolved in 100 μl 50% acetonitrile and the supernatant after centrifugation was used as samples, and 2 μl was injected for the detection of galactose in LC-MS. Chromatographic separation was performed on ACQUITY UPLC BEH Amide 1.7 μm 2.1 × 100 mm Column (waters, 186004801) with Van Guard precolumn (186004799) at 35˚C using an Acquity UPLC H-Class System (Waters). Mobile phases were as follows: Solvent A, 1% ammonia in 80% acetonitrile and solvent B, 1% (v/v) ammonia in 30% (v/v) acetonitrile. The gradient program was linear 0% to 60% A for 0 to 10 min at 0.2 ml/min flow rate. The ionized compounds were detected using XEVO TQ-S triple quadrupole tandem mass spectrometer coupled with electrospray ionization source (Waters). ESI source was positive-ion mode and MRM transitions were m/z 198.097 > 60.9. A standard curve was prepared based on the peak area using D-Galactose (Wako, 073–00031) and the concentration of each sample was calculated from the peak area using MassLynx 4.1 software (Waters). These data were normalized by the protein amount. The insoluble pellets that were extracted after removing the liquid phase were heat denatured with 0.2 N NaOH and used to quantify total protein with a BCA protein assay kit (Thermo Fisher).

## Statistics

Statistical tests used were shown in the figure legends. Graphpad Prism 9 was used for statistical analyses and plotting. Sample sizes were determined empirically based on the observed effects.

## Supporting information

**S1 Fig. Analysis of GALDAR in various conditions.** (A) Survival of adult male *OregonR* flies fed without (black line, *n* = 62), or with 5% galactose (red line, *n* = 76) addition to a basal agar medium containing 0.5% sucrose. Logrank (Mantel–Cox) test. (B) Confocal images of R2 regions of midguts of male *G2, UAS-GFP* flies fed with 0% (top) or 5% (bottom) galactose addition for 3d. ISCs are highlighted in small panels. (C) Confocal images of R2 regions of midguts of virgin (top) or mated (bottom) female *G1, tub-Gal4, UAS-GFP* flies fed with 0% or 5% galactose addition for 3d. ISCs are highlighted in small panels. (D) Confocal images of R2 regions of *act-Gal4* (top) or *tub-Gal4* (bottom) midguts with *UAS-GFP*. EECs are highlighted in small panels. Note the lack of GFP in *act-Gal4, UAS-GFP* EECs. (E) Confocal images of

midgut R2 regions of G2, virgin female *G2, UAS-nlsRFP* flies fed with 5% sucrose, 5% glucose, 30% glucose, and 5% galactose (DAPI in green, nlsRFP in magenta). (F) Quantification of nlsRFP fluorescence intensity per field of view ($n$ = 5 for each condition), ANOVA with multiple comparison test. Data are representative of at least 3 independent experiments. S1 Data provides the source data used for all graphs and statistical analyses. Scale bars, 20 μm in large, 5 μm in small panels in all figures.
(TIF)

**S2 Fig. Detailed analysis of galactose feeding and injection.** (A) Stacked confocal images of hindguts and abdomens (heart: between white lines) of adult virgin female G2, UAS-GFP flies with 1 day (left column), 2 days (middle column), and 3 days (right column) feeding with 0% (top row), 1% (middle row), or 5% (bottom row) galactose. (B) Quantification of mean GFP fluorescence intensities of heart, hindgut, and fat body shown in (A). Black lines show the mean value. Two-tailed unpaired $t$ test. (C) Single confocal images of adult virgin female *G2, UAS-GFP* flies 3 days after feeding (left) and 1 day after 20% galactose injection (right). Fluorescence intensity profiles of yellow lines indicated on the images (bottom), showing GFP signals in visceral muscles and ECs in hindguts after feeding and injection (DAPI in magenta, GFP in green). S1 Data provides the source data used for all graphs and statistical analyses. Scale bars, 50 μm in all figures.
(TIF)

**S3 Fig. Galactose metabolism in *Drosophila*.** (A) Confocal images of R2 regions of virgin adult female midguts showing *UAS-nlsRFP* only (top) or *Galk-Gal4, UAS-nlsRFP* (bottom) expression. ISCs are highlighted in small panels. (B) Stacked confocal images of virgin adult female abdomen, showing *UAS-nlsRFP* only (top) or *Galk-Gal4, UAS-nlsRFP* (bottom) expression in fat body, oenocytes, and heart muscle (between white lines). (C) Stacked confocal images of virgin adult female hindguts, showing *UAS-nlsRFP* only (top) or *Galk-Gal4, UAS-nlsRFP* (bottom) expression. (D) Survival of adult male $Galk^{-/-}$ and $Galk^{-/+}$ flies on 0% and 5% galactose supplemented food. Logrank (Mantel–Cox) test. (E) Confocal images of same midguts in live (top) and fixed condition (2-NBDG in green, DAPI in blue, Armadillo/Prospero in magenta). (F) Heatmap showing percentage of cells with detected expression of sugar transporters in cell clusters in *Drosophila* midgut according to single-cell RNA sequencing data. Note the lack of expression in ISC/EB.Data are representative of at least 3 independent experiments. S1 Data provides the source data used for all graphs and statistical analyses. Scale bars, 20 μm in large, 5 μm in small panels in (C), 50 μm in (D) and (E).
(TIF)

**S4 Fig. Expression of galactose metabolism genes and glycolysis genes in enterocytes.** (A) Heatmap showing percentage of cells with detected expression of galactose metabolism genes in enterocytes according to single-cell RNA sequencing data (aEC and pEC indicate ECs in the anterior and posterior parts, respectively). (B) Heatmap showing percentage of cells with detected expression of glycolysis genes in enterocytes according to single-cell RNA sequencing data (aEC and pEC indicate ECs in the anterior and posterior parts, respectively).
(TIF)

**S1 Data. The numerical values used for all main and supporting figures are provided.**
(XLSX)

## Acknowledgments

We thank Iswar Hariharan and the Bloomington Stock Center for fly stocks.

## Author Contributions

**Conceptualization:** Uğurcan Sakizli, Sa Kan Yoo.

**Formal analysis:** Uğurcan Sakizli, Sa Kan Yoo.

**Funding acquisition:** Sa Kan Yoo.

**Investigation:** Uğurcan Sakizli, Tomomi Takano, Sa Kan Yoo.

**Methodology:** Uğurcan Sakizli, Tomomi Takano, Sa Kan Yoo.

**Project administration:** Sa Kan Yoo.

**Supervision:** Sa Kan Yoo.

**Validation:** Uğurcan Sakizli.

**Visualization:** Uğurcan Sakizli.

**Writing – original draft:** Uğurcan Sakizli, Sa Kan Yoo.

**Writing – review & editing:** Uğurcan Sakizli, Tomomi Takano, Sa Kan Yoo.

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
