## [Editor Report · Decision Letter 0]

10 Jan 2024

Dear Dr Yoo, 

Thank you for submitting your revised manuscript entitled "GALDAR: a genetically encoded galactose sensor for visualizing sugar metabolism in vivo" for consideration as a Methods and Resources article by PLOS Biology. I have now had a chance to discuss your revision with the academic editor and I am writing to let you know that we would like to send your submission back to the original reviewers.

Once your full submission is complete, your paper will undergo a series of checks in preparation for peer review. After your manuscript has passed the checks it will be sent out for review. To provide the metadata for your submission, please Login to Editorial Manager (https://www.editorialmanager.com/pbiology) within two working days, i.e. by Jan 12 2024 11:59PM.

Kind regards,

Luke

Lucas Smith, Ph.D.

Senior Editor

PLOS Biology

lsmith@plos.org

---

## [Decision Letter · Decision Letter 1]

1 Feb 2024

Dear Sakan,

Thank you for your patience while we considered your revised manuscript "GALDAR: a genetically encoded galactose sensor for visualizing sugar metabolism in vivo" as a Methods and Resources at PLOS Biology. Your revised study has now been evaluated by the PLOS Biology editors, the Academic Editor and the original reviewers. 

As you will see, below, reviewers 2 and 3 are fully satisfied by the revision, and reviewer 1 agrees that the study has been strengthened. However, reviewer 1 also notes a few places where the writing might be improved, and suggests that you perform a last bit of analyses to help clarify the regional expression patterns of genes encoding galactose catabolizing enzymes. 

In light of the reviews, which you will find at the end of this email, we are pleased to offer you the opportunity to address the remaining points from the reviewer 1, in a revision that we anticipate should not take you very long. We will then assess your revised manuscript and your response to the reviewers' comments with our Academic Editor aiming to avoid further rounds of peer-review, although might need to consult with the reviewers, depending on the nature of the revisions.

**IMPORTANT: As you address the last reviewer comments, we also ask that you attend to the following editorial requests: 

1) FINANCIAL DISCLOSURES: Please update your financial disclosures statement, in our online system, to describe the role of any sponsors or funders in the study design, data collection and analysis, decision to publish, or preparation of the manuscript. If the funders had no role in any of the above, include this sentence at the end of your statement: "The funders had no role in study design, data collection and analysis, decision to publish, or preparation of the manuscript."

2) FLIES: For the benefit of the scientific community, we encourage you to deposit the GALDAR flies in a public stock center such as VDRC (Vienna) or BDSC (Bloomington), as we think this will facilitate the widespread use of these flies. 

3) CODE: Per journal policy, if any code was generated that is important to support the conclusions of your manuscript, we would require that you make it available without restrictions upon publication. Please ensure that any code is sufficiently well documented and reusable, and that your Data Statement in the Editorial Manager submission system accurately describes where your code can be found.

4) DATA: You may be aware of the PLOS Data Policy, which requires that all data be made available without restriction: http://journals.plos.org/plosbiology/s/data-availability. For more information, please also see this editorial: http://dx.doi.org/10.1371/journal.pbio.1001797

a. Supplementary files (e.g., excel). Please ensure that all data files are uploaded as 'Supporting Information' and are invariably referred to (in the manuscript, figure legends, and the Description field when uploading your files) using the following format verbatim: S1 Data, S2 Data, etc. Multiple panels of a single or even several figures can be included as multiple sheets in one excel file that is saved using exactly the following convention: S1_Data.xlsx (using an underscore).

b. Deposition in a publicly available repository. Please also provide the accession code or a reviewer link so that we may view your data before publication. 

>>>Regardless of the method selected, please ensure that you provide the individual numerical values that underlie the summary data displayed in the following figure panels as they are essential for readers to assess your analysis and to reproduce it:

Figure S1A,F; Fig 1F-G; Fig2D; Fig S2B; Fig 3C-D; Fig S3D; Fig 5D-E; Fig6C,E,G;

>>Please also ensure that *figure legends* in your manuscript include information on where the underlying data can be found, and ensure your supplemental data file/s has a legend.

>>Please ensure that your Data Statement in the submission system accurately describes where your data can be found.

**IMPORTANT - SUBMITTING YOUR REVISION**

*Resubmission Checklist*

*Published Peer Review*

*Blot and Gel Data Policy*

Sincerely,

Luke

Lucas Smith, Ph.D.

Senior Editor

PLOS Biology

lsmith@plos.org

REVIEWS:

Reviewer #1: The revisions have addressed many of my original concerns, including GALDAR incompatibility with GAL4/UAS mediated gene manipulation, lacking evidence for the detection of physiologically relevant galactose levels, and the insufficient introduction of the physiological relevance of galactose metabolism. The addition of GALDAR3 data is a significant improvement. While the proof of principle, i.e. the demonstration of the potential of the tool to advance biological understanding, still remains pretty limited, I am inclined to recommend the manuscript to be accepted, pending the revisions outlined below.

Specific concerns: 

1. The text has been somewhat improved and the erroneous statements pointed out earlier have been eliminated. There are still many paragraphs where the quality of writing is below the standard usually seen in journals like Plos Biology. Some examples:

-"…the fly's ancestral fruit, marula contains galactose…"

-"We heterologously transplanted yeast's galactose sensing system into Drosophila. Yeast senses galactose using Gal4, Gal80 and Gal3. …. We implanted yeast's whole Gal system to Drosophila and generated a heterologous biosensor for galactose. …. We transgenically incorporated the yeast galactose metabolism genes, including Gal3, Gal4 and Gal80…"

"We have thus far detected galactose by GALDAR in adult flies. We investigated galactose metabolism during larval development. During larval development, there is an ecdysteroid-mediated dynamic change of sugar metabolism [35-37]. GALDAR larvae were developed with 0% and 5% galactose (Fig. 5A)…"

Notably, the new text included in the end of the Results section to describe GALDAR3 is written in a more proficient way. The rest of the text should be carefully edited to meet the same standards.

2. I earlier made a comment about the detected GALDAR sensor activity in the anterior midgut, which does not seem to reflect high, but rather low, galactose metabolism in the anterior midgut (leading to galactose accumulation). This is because inhibition of galactose metabolism increases the signal in the posterior part. Now the authors state the following (in two separate parts):

"We observed that the anterior midgut (R2 region) of adult virgin females exhibits GFP signals with a presence of galactose (Fig 1D). This observation is reminiscent of the previous reports showing that carbohydrate metabolism is compartmentalized in the anterior midgut of female Drosophila [31, 32]."

"On the Galk mutant background, most tissues and cell types were GALDAR positive after 2-day feeding with 2% galactose (Fig 4B). This suggests that galactose is efficiently metabolized in most tissues at a relatively high rate, which affects the intracellular concentrations of galactose detected by GALDAR. Consistently, the whole adult midgut exhibits GALDAR signals after feeding 2% galactose with the Galk mutant background as opposed to the restricted GALDAR signals in anterior midgut with 5% galactose feeding in the wildtype background (Fig 4C)."

The data shown in Figure 1 (when considered alone) does indeed suggest that also galactose metabolism occurs in the anterior parts. However, when taken into account the data in figure 4, the posterior part seems in fact to be faster in catabolizing Galactose than the anterior part. This is in contrast to the earlier findings (on anterior expression of carbohydrate metabolism genes). To clarify the issue, it would be advisable to relate the findings of this manuscript to the existing data in a more specific and explicit manner. Rather than looking at random carbohydrate metabolism genes unrelated to galactose, to use the existing published datasets and specifically analyze the expression of genes encoding galactose catabolizing enzymes in the anterior vs. posterior parts of the intestine. This would allow you to answer whether their regional expression corresponds to the data produced by the GALDAR sensor.

Reviewer #2: My comments have been addressed.

Reviewer #3: The authors have addressed my concerns

---

## [Editor Report · Decision Letter 2]

14 Feb 2024

Dear Sakan,

Thank you for the submission of your revised Methods and Resources article "GALDAR: a genetically encoded galactose sensor for visualizing sugar metabolism in vivo" for publication in PLOS Biology and thank you for addressing the last editorial and reviewer requests in this revision. On behalf of my colleagues and the Academic Editor, Alex P Gould, I am pleased to say that we can in principle accept your manuscript for publication, provided you address any remaining formatting and reporting issues. These will be detailed in an email you should receive within 2-3 business days from our colleagues in the journal operations team; no action is required from you until then. Please note that we will not be able to formally accept your manuscript and schedule it for publication until you have completed any requested changes.

**IMPORTANT: As you address any formatting requests to come we ask that you also address the following editorial requests: 

1) DATA: thank you for providing the underlying data for your study as a supplemental file. Please add a brief sentence directing readers to this file to each relevant figure legend (including supplemental). For example, you can add the sentence "the data underlying this figure can be found in S1_Data"

2) We would like to suggest a couple of word choice changes in some of the descriptions in the study for clarity. Specifically, we think in your new responses the words "transplanted"/"implanted" should be replaced with "transferred" or "incorporated" as we think this is a bit clearer. 

PRESS

Sincerely, 

Luke

Lucas Smith, Ph.D.

Senior Editor

PLOS Biology

lsmith@plos.org